



# Downpour Dynamics: Outsized impacts of storm events
# on unprocessed atmospheric nitrate export in an urban
# watershed
Joel T. Bostic[1,2], David M. Nelson[1], Keith N. Eshleman[1]
[1]University of Maryland Center for Environmental Science, Appalachian Lab, Frostburg, Maryland,
USA
[2]Potomac State College of West Virginia University, Keyser, West Virginia, USA
*Correspondence to*: Joel Bostic (joel.bostic@mail.wvu.edu)
**Abstract.** Water-quality impacts of streamwater nitrate ($NO_3^-$) on downstream ecosystems are largely
determined by the load of $NO_3^-$ from the watershed to surface waters. The largest $NO_3^-$ loads often
occur during storm events, but it is unclear how loads of different $NO_3^-$ sources change during storm
events relative to baseflow or how watershed attributes might affect source export. To assess the role
of stormflow and baseflow on $NO_3^-$ source export and how these roles are modulated by hydrologic
effects of land-use practices, we measured nitrogen ($\delta^{15}N$) and triple oxygen ($\Delta^{17}O$) isotopes of $NO_3^-$
and oxygen isotopes ($\delta^{18}O$) of water in rainfall and streamwater samples from before, during, and after
8 storm events across 14 months in two Chesapeake Bay watersheds of contrasting land-use. Storms
had a disproportionately large influence on the export of unprocessed atmospheric $NO_3^-$ ($NO_3^-{}_{Atm}$) and
a disproportionately small influence on export of terrestrial $NO_3^-$ ($NO_3^-{}_{Terr}$) relative to baseflow in the
developed urban watershed. In contrast, baseflow and stormflow had similar influences on $NO_3^-{}_{Atm}$
and $NO_3^-{}_{Terr}$ export in the mixed agricultural/forested watershed. An equivalent relationship between
$NO_3^-{}_{Atm}$ deposition on impervious surfaces and event $NO_3^-{}_{Atm}$ streamwater export in the urban
watershed suggests that impervious surfaces that hydrologically connect runoff to channels likely
facilitate export of $NO_3^-{}_{Atm}$ during rainfall events. Additionally, larger rainfall events were more
effective in exporting $NO_3^-{}_{Atm}$ in the urban watershed, with increased rainfall depth resulting in a
greater fraction of event $NO_3^-{}_{Atm}$ deposition exported. Considering both projected increases in
precipitation amounts and intensity and urban/suburban sprawl in many regions of the world, best
management practices that reduce hydrologic connectivity of impervious surfaces will likely help to
mitigate the impact of storm events on $NO_3^-{}_{Atm}$ export from developed watersheds.

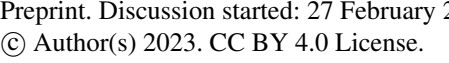



## 1 Introduction

Increasing streamwater nitrate ($NO_3^-$) export over the past century has negatively impacted

many downstream ecosystems globally (Kemp et al., 2005; Camargo and Alonso, 2006; Steffen et al.,
2015; Stevens, 2019). The severity of impacts to receiving waters is partially determined by the
magnitude of $NO_3^-$ loads (i.e., product of concentration and discharge; NRC, 2000). As such, riverine
$NO_3^-$ loads are greatest during periods of high discharge, which often follow large precipitation
events, and can therefore have an outsized impact on annual streamwater $NO_3^-$ loads (Vaughan et al.,
2017; Kincaid et al., 2020). Sources of $NO_3^-$ comprising storm event loads can be variable and
associated with changing hydrologic flowpaths during precipitation events (Buda and DeWalle, 2009).
Loads of individual $NO_3^-$ sources (e.g., atmospheric $NO_3^-$) exported during storm events are rarely
quantified, however (Divers et al., 2014; Sabo et al., 2016). Thus, it is not clear whether storm events
have a disproportionate impact relative to non-storm (i.e., baseflow) conditions on different $NO_3^-$
sources. The impact of storm events relative to baseflow on sources of streamwater $NO_3^-$ is
particularly relevant given the increases in precipitation amount and intensity projected to be
associated with future climate change (Walsh et al., 2014).

Precipitation can affect the amount, as well as the source, of $NO_3^-$ exported in surface waters

via the surface-to-stream flow path. During storms, $NO_3^-$ can be transported to streams by either
overland or subsurface pathways. Overland flow is associated with $NO_3^-$ sources deposited or present
on the land surface, such as unprocessed atmospheric $NO_3^-$ ($NO_3^-_{Atm}$; Rose et al., 2015a). Subsurface
flow is associated with $NO_3^-$ sources abundant in soils and groundwater, such as fertilizer, microbial,
and/or sewage (Cook and Herczeg, 2012). Both hydrologic flowpaths (and the respective $NO_3^-$
sources) can be affected by human land-use activities (Paul and Meyer, 2001; Barnes and Raymond,
2010; Jarvis, 2020). For example, previous studies report that developed watersheds export relatively
more $NO_3^-_{Atm}$ than less developed watersheds, presumably due to hydrologic changes created by
impervious surfaces (Buda and DeWalle, 2009; Burns et al., 2009; Kaushal et al., 2011; Bostic et al.,
2021). However, evidence is lacking for (1) the mechanism generating increased $NO_3^-_{Atm}$ export in





developed watersheds and (2) quantitative impacts of storm event loads relative to baseflow, both of
which could be useful for mitigating the effects of storms on streamwater $NO_3^-$ export.

The stable isotope compositions of $NO_3^-$ and water ($H_2O$) are powerful tools for

distinguishing $NO_3^-$ sources and hydrologic flow paths, respectively. For example, the triple oxygen
isotope values ($\Delta^{17}O$) of $NO_3^-$ allow for quantification of atmospheric and terrestrial sources of $NO_3^-$
in streamwater (Michalski et al., 2003), and $\delta^{15}N$ and $\delta^{18}O$ values of $NO_3^-$ permit inferences into the
relative contributions of terrestrially-sourced $NO_3^-$ ($NO_3^-{}_{Terr}$), such as fertilizer or sewage N (Kendall
et al., 2007). Additionally, $\delta^{18}O$ values of $H_2O$ can be used to assess the importance of overland versus
subsurface flow through partitioning of stream flow into pre-event and event contributions (Sklash et
al., 1976; McGuire and McDonnell, 2007). Few studies have coupled these isotopic tracers (Buda and
DeWalle, 2009), however, despite their suitability to assess the effect of storm events on both
hydrologic flow paths and export of different $NO_3^-$ sources. Such information could provide
mechanistic evidence for the commonly reported relationship between developed watersheds and
$NO_3^-{}_{Atm}$ export.

Here we address the following research questions: How do storm events affect the total

amount and sources of $NO_3^-$ exported in streams relative to baseflow? And, more specifically, what is
the relationship between hydrologic and biogeochemical effects of land use and the export of
unprocessed atmospheric $NO_3^-{}_{Atm}$ and terrestrial $NO_3^-$ during storm events and baseflow? These
questions were addressed in two Chesapeake Bay watersheds of contrasting land-use. A two-watershed
study is inherently comparative, potentially limiting the inferences that can be made regarding land-use
effects. However, given the contrasting land uses (i.e., predominantly developed compared to mixed
forest/agriculture) in these watersheds, we believe that this study can adequately address our research
questions while presenting a "proof of concept" for future studies. To address these research questions,
we collected moderate-frequency streamwater samples before, during, and after eight rainfall events,
bulk rainfall samples corresponding to these events, as well as monthly baseflow samples, in two
catchments within the broader Chesapeake Bay watershed. We then used $\delta^{15}N$, $\delta^{18}O$, and $\Delta^{17}O$ of
$NO_3^-$ and $\delta^{18}O$ of $H_2O$ to determine $NO_3^-$ sources and hydrologic flowpaths, respectively. The
Chesapeake Bay region is ideal for our study: it is one of the most ecologically and economically



important estuaries in the world (NOAA, 1990) that has experienced recent improvements in
ecosystem health associated with declining N loads (Chanat et al., 2016; Lefcheck et al., 2018; Zhang
et al., 2018), but uncertainty surrounds continued water quality improvements in part due to the effects
of projected increases in precipitation intensity across its diverse land-use watershed (Najjar et al.,

2010).

## 2 Materials and Methods

### 2.1 Study watersheds and field methods

To assess $NO_3^-$ export dynamics during storm events, streamwater and rainfall samples were
collected synchronously during eight events from two watersheds with outlets in Maryland, USA –
Gwynns Falls at Villa Nova (GWN) and Gunpowder Falls at Glencoe (GUN) (Figure 1) – from
September 2018 – October 2019. These watersheds have similar geology (Piedmont physiographic
province; Fenneman, 1946) and climate (humid sub-tropical; Kottek et al., 2006), but differing land-
use (one predominantly developed and the other mixed forest and agriculture), impervious surface
coverage (Figure S1) and area (Table 1). Events were targeted based on forecast precipitation amounts
of at least 2.5 cm and the same events were sampled at each site. Automated samplers (Teledyne ISCO
3700 Portable Sampler, Lincoln, NE) were used to collect streamwater samples into pre-cleaned 1L
bottles across each storm hydrograph, including pre-storm baseflow, rising limbs, and falling limbs for
most events at intervals ranging from 45 minutes – 12 hours (Figure S2). Storm sample collection
ceased when discharge fell to approximately 200% of pre-event baseflow. Bulk rainfall samples
corresponding to each event were collected using 7.5 cm diameter funnels approximately 1 m above
ground level connected to pre-cleaned 1 L Nalgene bottles, with pre-cleaned table tennis balls used to
limit evaporation. Streamwater and rainfall samples were placed on ice for 12 – 36 hours after
collection, then processed in the laboratory within 24 – 48 hours. Both study watersheds are gaged by
the United States Geological Survey; 15-minute and mean daily discharge data were obtained using the
dataRetrieval R package (DeCicco, 2018). Mean event rainfall depth for each watershed was obtained
from PRISM Climate Group (PRISM, 2014) using the prism R package (Hart and Bell, 2015).





## 2.2 Lab Methods

Streamwater and rainfall samples for $NO_3^-$ concentration and isotope analyses were filtered (0.45 μm) and frozen within 48 hours of collection. Aliquots for water isotope measurements were stored in completely filled (i.e., no headspace) 20 mL bottles at room temperature prior to analysis. $NO_3^-$ and nitrite ($NO_2^-$) concentrations were measured using flow-injection colorimetric analysis (Lachat Quikchem 8000 FIA+).

The $\Delta^{17}O$, $\delta^{18}O$, and $\delta^{15}N$ values of stream and rainfall $NO_3^-$ were measured using a Thermo Delta V+ isotope ratio mass spectrometer (Bremen, Germany) via the denitrifier method (Sigman et al., 2001; Casciotti et al., 2002) with thermal decomposition (at 800° C) of $N_2O$ to $N_2$ and $O_2$ (Kaiser et al., 2007) at the Central Appalachians Stable Isotope Facility. $NO_2^-$ is denitrified using this method as well, but $NO_2^-$ concentrations in stream and rainfall samples were low relative to $NO_3^-$ ($NO_2^-$/($NO_2^-$ + $NO_3^-$) mean = 0.006, range = 0.00 – 0.027). Measured isotope ratios were normalized using international reference standards USGS 34 ($\delta^{17}O$ = -14.8 ‰, $\delta^{18}O$ = -27.9 ‰) and USGS 35 ($\delta^{17}O$ = 51.5 ‰, $\delta^{18}O$ = 57.5 ‰) for O isotopes (Böhlke et al., 2003) and USGS 32 ($\delta^{15}N$ = 180 ‰) and USGS 34 ($\delta^{15}N$ = -1.8 ‰) for N isotopes (IAEA, 1995). Reference standards were measured throughout sample analysis in equal concentrations to samples (ranging from 100 – 200 nmol depending on sample $NO_3^-$ concentration). Analytical precision of $\Delta^{17}O$ ($\Delta^{17}O \approx \delta^{17}O - 0.52 \times \delta^{18}O$) was estimated as 0.5 ‰, $\delta^{18}O$ as 1.4 ‰, and $\delta^{15}N$ as 1.8 ‰ (1 σ), based on repeated measurements (n ≅ 200) of reference standards USGS 32 and USGS 35 and a laboratory reference standard "Chile $NO_3^-$" (Duda Energy 1sn 1 lb. Sodium Nitrate Fertilizer 99+% Pure Chile Saltpeter from Amazon.com). Accuracy of $\Delta^{17}O$, $\delta^{18}O$, and $\delta^{15}N$ were tracked using repeated measurements of IAEA-N3 (n = 19, mean $\Delta^{17}O$ = -0.1 ‰, $\delta^{18}O$ = 24.3 ‰, $\delta^{15}N$ = 4.5 ‰) and closely agreed with published values (IAEA, 1995; Michalski et al., 2002; Böhlke et al., 2003). Each streamwater and rainfall sample was measured 3 – 6 times to reduce analytical uncertainty and the mean of each sample was used in all analyses. Standard error of the mean ranged from 0.1 – 0.6 ‰, 0.1 – 1.6 ‰, and 0.1 – 1.6 ‰ for replicate measurements of $\Delta^{17}O$, $\delta^{18}O$, and $\delta^{15}N$ respectively.





Oxygen ($\delta^{18}O_{\text{-H2O}}$) isotopes of rainfall and streamwater were measured using a Picarro L2130-

i via cavity ring down spectroscopy at the University of Wyoming Stable Isotope Facility. Measured

isotope ratios were normalized to VSMOW using internal laboratory standards that were calibrated to

international standards. Precision based on repeated measurements of internal standards was 0.2 ‰.

**2.3 Quantification of atmospheric $NO_3^-$ deposition**

Event $NO_3^-{}_{\text{Atm}}$ deposition was quantified using the measured rainfall $NO_3^-$ concentration and

mean rainfall depth:

$NO_{3-Atm}^- Deposition\ (g\ N\ ha^{-1}) = \dfrac{Rainfall\ Volume\ (L)\times Rainfall\ NO_3^-\ (mg\ N\ L^{-1})}{Watershed\ Area\ (ha)} \times (1\times 10^{-3})$ (eq.

1)

where rainfall volume is the product of rainfall depth and watershed area and $1\times 10^{-3}$ is a conversion

factor. Event $NO_3^-{}_{\text{Atm}}$ deposition onto impervious surfaces was then calculated by multiplying $NO_3^-{}_{\text{Atm}}$

deposition by the percent of impervious surfaces.

**2.4 Quantification of unprocessed atmospheric and terrestrial $NO_3^-$ in streams**

Concentrations of $NO_3^-{}_{\text{Atm}}$ were quantified using $\Delta^{17}O$ values of terrestrial and rainfall end-

members and total $NO_3^-$ concentrations:

$f_{Atm} = \dfrac{(\Delta^{17}O_{Stream} - \Delta^{17}O_{Terr})}{(\Delta^{17}O_{Precip} - \Delta^{17}O_{Terr})}$        (eq. 2)

$NO_{3Atm}^-(mg\ N\ L^{-1}) = f_{Atm} \times NO_{3Total}^-(mg\ N\ L^{-1})$        (eq. 3)

$NO_{3Terr}^-(mg\ N\ L^{-1}) = NO_{3Total}^-(mg\ N\ L^{-1}) - NO_{3Atm}^-(mg\ N\ L^{-1})$        (eq. 4)

where $\Delta^{17}O_{\text{Stream}}$ = $\Delta^{17}O$ of streamwater samples during either baseflow or storm events, $\Delta^{17}O_{\text{Precip}}$ =

$\Delta^{17}O$ of rainfall for a given event, $\Delta^{17}O_{\text{Terr}}$ = $\Delta^{17}O$ of terrestrially sourced $NO_3^-$ which is $\cong 0$ ‰,

$NO_3^-{}_{\text{Terr}}$ = terrestrial $NO_3^-$, and $NO_3^-{}_{\text{Total}}$ = measured streamwater $NO_3^-$ concentrations. Uncertainty

in $NO_3^-{}_{\text{Atm}}$ was estimated by propagating analytical uncertainty from repeated measures of $\Delta^{17}O_{\text{Stream}}$

and $\Delta^{17}O_{\text{Precip}}$.

**2.5 Quantification of event loads and mean concentrations and monthly loads**

Event loads of $NO_3^-{}_{\text{Total}}$ and $NO_3^-{}_{\text{Atm}}$ were calculated as:



$$L_{NO_3^-} = \sum_{i=1}^{n} C_i \times V_i \times (1 \times 10^{-3}) \qquad \text{(eq. 5)}$$

where $L$ = load of either $NO_3^-{}_{Total}$, $NO_3^-{}_{Atm}$, or $NO_3^-{}_{Terr}$ in g per event, $C_i$ = concentration of either

$NO_3^-{}_{Total}$ or $NO_3^-{}_{Atm}$ in mg N L$^{-1}$ for sample $i$, and $V_i$ = volume of water exported corresponding to

sample $i$ in L, and $1 \times 10^{-3}$ is a conversion factor (mg to g). Event yields (g N ha$^{-1}$ event$^{-1}$) of $NO_3^-{}_{Total}$,

$NO_3^-{}_{Atm}$, and $NO_3^-{}_{Terr}$ were calculated by normalizing loads by watershed area. To assess potential

bias between our method (eq. 5) and traditionally used methods to quantify $NO_3^-{}_{Atm}$, we used the mean

daily discharge multiplied by $NO_3^-{}_{Atm}$ concentrations of each individual grab sample collected during

a particular event. We compared these estimated loads with the "true" load (calculated using eq. 5) and

calculated bias as the difference between the "true" load and loads estimated using a single sample and

daily average discharge. Because traditional methods commonly use mean daily discharge, we only

investigated bias for two events that included samples collected over one full day. We also calculated

the event fraction of unprocessed atmospheric $NO_3^-$ ($f_{Atm}$) using $\Delta^{17}O$ (eq. 2) and $\delta^{18}O$ (substituting

$\delta^{18}O$ for $\Delta^{17}O$ in eq. 2 and assuming that baseflow samples for a corresponding storm represent the

terrestrial $NO_3^-$ end-member $\delta^{18}O$ value).

Event mean concentrations (EMC) of $NO_3^-{}_{Total}$ and $NO_3^-{}_{Atm}$ and event mean values (EMV)

of $\Delta^{17}O$, $\delta^{18}O$, and $\delta^{15}N$ were calculated as:

$$EMC, EMV = \frac{\sum_{i=1}^{n}(C_i \times V_i)}{\sum_{i=1}^{n} V_i} \qquad \text{(eq. 6)}$$

where EMC = event mean concentration in mg N L$^{-1}$ (for $NO_3^-{}_{Total}$ and $NO_3^-{}_{Atm}$), EMV = event mean

value in ‰ ($\Delta^{17}O$, $\delta^{18}O$, and $\delta^{15}N$), $C_i$ = either concentration of $NO_3^-{}_{Total}$ or $NO_3^-{}_{Atm}$ (mg N L$^{-1}$) or

value of $\Delta^{17}O$, $\delta^{18}O$, or $\delta^{15}N$ (‰) corresponding to sample $i$, and $V_i$ = volume of water exported

corresponding to sample $i$ (L).

Monthly loads of $NO_3^-{}_{Total}$ were estimated using Weighted Regressions on Time, Discharge,

and Season Kalman Filter (WRTDS-K; Zhang and Hirsch, 2019). Regressions were calibrated using

the entire period of record for $NO_3^-$ (excluding our storm samples) to generate coefficients

representing a greater range of hydroclimatological conditions than was realized in 13 months. $NO_3^-$

concentration data for the entire period of record were obtained from the Chesapeake Bay Program

water quality database (Chesapeake Bay Program, 2021). Our storm samples were excluded to





generate similar estimates of monthly and annual loads used by monitoring agencies (e.g., Maryland
Department of Natural Resources, US Environmental Protection Agency) in these watersheds.
Monthly yields (g N ha$^{-1}$) were calculated by dividing monthly loads by watershed area and monthly
flow-weighted concentrations (mg N L$^{-1}$) were calculated by dividing monthly loads by monthly
discharge. Uncertainty of $NO_3^-{}_{Total}$ was estimated using block bootstrapping methods for WRTDS-K
(Zhang and Hirsch, 2019) and was propagated through all analyses using $NO_3^-{}_{Total}$ loads and/or yields.
**2.6 Terrestrial δ$^{18}$O and δ$^{15}$N calculation**
Streamwater storm samples of $\delta^{18}$O and $\delta^{15}$N were corrected to remove the influence of
$NO_3^-{}_{Atm}$ (Dejwakh et al., 2012), which has higher $\delta^{18}$O values and can have lower $\delta^{15}$N values than
terrestrial $NO_3^-$ (Elliott et al., 2007; Kendall et al., 2007). This was done to more carefully infer how
terrestrial sources of $NO_3^-$ might change during storm events, and it uses the following equations:
$$\delta^{15}N_{Terr} = \frac{(\delta^{15}N_{Stream} - \delta^{15}N_{Atm} \times f_{Atm})}{f_{Terr}} \quad \text{(eq. 7)}$$

$$\delta^{18}O_{Terr} = \frac{(\delta^{18}O_{Stream} - \delta^{18}O_{Atm} \times f_{Atm})}{f_{Terr}}$$

$$\text{(eq. 8)}$$

where $\delta^{15}$N/$\delta^{18}$O$_{Stream}$ = measured $\delta^{15}$N or $\delta^{18}$O of streamwater storm samples, $\delta^{15}$N/$\delta^{18}$O$_{Atm}$ = rainfall
$\delta^{15}$N or $\delta^{18}$O for a given event, $f_{Atm}$ = fraction of $NO_3^-{}_{Atm}$, as calculated using eq. 2, and $f_{Terr}$ = 1- $f_{Atm}$.
**2.7 Hydrograph separation**
Water isotopes were used to quantify the proportion of event and pre-event water during
storm events at or near peak discharge. The direct routing, or translation of rainfall to streamwater
during the same event, was quantified as the event-water fraction (i.e., rainfall), whereas water present
in the catchment prior to the storm event was classified as the pre-event water fraction (i.e., baseflow)
using the following equations (Sklash et al., 1976):
$$f_{Event\ Water} + f_{Pre-Event\ Water} = 1 \quad \text{(eq. 9)}$$

$$f_{Event\ Water} = \frac{\delta^{18}O_{PeakQ} - \delta^{18}O_{Baseflow}}{\delta^{18}O_{Precipitation} - \delta^{18}O_{Baseflow}} \quad \text{(eq. 10)}$$



where $\delta^{18}O_{PeakQ}$ = $\delta^{18}O_{H2O}$ at or near peak discharge during storm events, $\delta^{18}O_{Baseflow}$ = $\delta^{18}O_{H2O}$ of
streamwater just prior to storm event and hydrograph rise, and $\delta^{18}O_{Rainfall}$ = $\delta^{18}O_{H2O}$ of bulk rainfall
samples during a given storm event. Event and pre-event water runoff can be quantified using these
equations by multiplying runoff during peak stormflow by fractions of event and pre-event water.
Uncertainty was estimated using published methods to account for analytical uncertainty and
separation, or lack thereof, of end-members (Genereux, 1998). It has been shown that some of the
assumptions of isotope-based hydrograph separation may be violated in mesoscale catchments (e.g.,
spatiotemporally constant end-member values; Klaus and McDonnell, 2013), thus we estimate event-
water fractions and runoff for peak discharge only and apply these data cautiously.
**2.8 Framework for interpreting baseflow and stormflow contributions**
The importance of storm events relative to baseflow in streamwater $NO_3^-$ export can be
evaluated using a fractional export plot (Figure 2). In this plot the y-axis shows the fraction of annual
nitrate loads exported during a single event ($f_{NO3}$) and the x-axis shows the fraction of annual discharge
exported during a single event ($f_{Runoff}$). For example, if $NO_3^-$ concentrations remain constant with
changing discharge during a storm, the data would fall on the 1:1 line because its load is perfectly
explained by discharge and both storm events and baseflow have equal impact on loads (Figure 2). If
$NO_3^-$ concentrations decrease with increasing discharge during a storm, the data would plot below the
1:1 line. Watersheds with events consistently plotting below the 1:1 line indicate that baseflow, relative
to storm events, has an outsized impact on riverine nitrate loads. If $NO_3^-$ concentrations increase with
increasing discharge, the data would plot above the 1:1 line. Watersheds with events consistently
plotting above the 1:1 line indicate that storm events have an outsized impact on riverine $NO_3^-$ loads.
This framework can be expanded further by quantifying the (potential) disproportionate effect of storm
events on streamwater constituent loads relative to water yields. Dividing $f_{NO3}$ by $f_{Runoff}$ provides a
single value to quantify the level of disproportionality:
$$Disproportionality\ Factor\ (DF) = \frac{f_{NO3}}{f_{Runoff}} \qquad (eq.\ 11)$$
$DF$ can be interpreted using Figure 2: a value falling on the 1:1 line would have $DF$ = 1, a value below
the 1:1 line would have a $DF$ < 1, and a value above the 1:1 line would have $DF$ > 1. For example, an



event with $DF = 4$ indicates that a given storm exported $4\times$ more $NO_3^-$ than water whereas an event
with $DF = 0.5$ indicates that a storm exported.
**2.9 Statistical analyses**
All statistical tests were performed in R (R Development Core Team, 2019). A Wilcoxon
ranked-sum test was used to compare EMC and EMV of paired streamwater storm and baseflow
samples. Due to the presence of outliers, Theil-Sen slopes (calculated using the *senth* function in R)
were used to assess relationships between most continuous variables (Helsel et al., 2020). Least
squares linear regression was used when outliers were absent. Confidence intervals (95%) and p-values
of Theil-Sen slopes were computed using bootstrapping (10,000 replicates) to incorporate uncertainty
in $DF$ and event-water fractions.
**3 Results**
Rainfall depth and chemistry ($NO_3^-$ concentrations and isotopes, $H_2O$ isotopes) were similar
between watersheds for sampled events ($p > 0.1$, Table S1). Rainfall depths ranged from $1.90 - 8.10$
cm, which corresponds to a range of 24-hour precipitation depth return intervals of <1 year (1-year
return interval $\approx 6.75$ cm) up to 2-year (2-year return interval $\approx 8.3$ cm) in this region (Bonnin et al.,
2004). Streamwater $NO_3^-$ concentrations ranged from $0.05 - 0.26$ mg N $L^{-1}$, $\delta^{15}N\text{-}NO_3^-$ from -8.7 – -
1.4 ‰, $\delta^{18}O\text{-}NO_3^-$ from $48.0 - 69.6$ ‰, and $\Delta^{17}O\text{-}NO_3^-$ from $13.6 - 24.9$ ‰. Streamflow was slightly
more variable in GWN during storm events (Table S2): event mean runoff and event maximum runoff
were higher in GWN ($p < 0.05$ and $p < 0.01$ respectively), but event median runoff was not different
between the watersheds ($p = 0.11$). Across all flow conditions, $NO_3^-$ concentrations were lower at
GWN (median = 0.78 mg N $L^{-1}$) than GUN (median = 2.60 mg N $L^{-1}$). Baseflow $NO_3^-$ concentrations
were higher than stormflow $NO_3^-$ EMCs in both watersheds, but differences were more pronounced at
GWN (baseflow median = 1.79 mg N $L^{-1}$, storm median = 0.66 mg N $L^{-1}$, $p < 0.05$) than GUN
(baseflow median = 3.06 mg N $L^{-1}$, storm median = 2.55 mg N $L^{-1}$, $p < 0.05$, Figure 3 and Table S3).
At GWN, values of $\delta^{15}N$ were higher in baseflow (median $\delta^{15}N$ = 7.6 ‰) than stormflow
(EMV median $\delta^{15}N$ = 5.0 ‰, respectively, $p < 0.05$), whereas values of $\delta^{18}O\text{-}NO_3^-$ were lower in
baseflow (median $\delta^{18}O$ = 3.9 ‰) than stormflow (EMV median $\delta^{18}O$ = 7.4 ‰, $p < 0.05$). In contrast,



values of $\delta^{15}$N- and $\delta^{18}$O-NO$_3^-$ did not differ between baseflow and stormflow at GUN (baseflow
median $\delta^{15}$N = 6.2 ‰, $\delta^{18}$O = 3.3 ‰; stormflow EMV median $\delta^{15}$N = 6.1 ‰, $\delta^{18}$O = 3.0 ‰; Figure 3
and Table S3). Values of $\delta^{18}$O-NO$_3^-{}_{Terr}$ were higher during baseflow at both sites (p < 0.05, Figure 3),
whereas $\delta^{15}$N-NO$_3^-{}_{Terr}$ was higher during baseflow at GWN only (p < 0.05, Figure 3). Similarly, $\Delta^{17}$O
of NO$_3^-$ was not significantly different between baseflow (median = 0.4 ‰) and stormflow (EMV
median = 0.5 ‰) at GUN, but was lower during baseflow (median = 0.7 ‰) than stormflow (EMV
median = 2.0 ‰, p < 0.05, Figure 3 and Table S3) at GWN.

Concentrations of NO$_3^-{}_{Terr}$ were more temporally variable than NO$_3^-{}_{Atm}$. Concentrations of

NO$_3^-{}_{Terr}$ showed similar patterns to NO$_3^-{}_{Total}$ at both watersheds: higher during baseflow than storm
events (GWN baseflow median = 1.72 mg N L$^{-1}$, stormflow median = 0.59 mg N L$^{-1}$; p < 0.001, GUN
baseflow median = 3.03 mg N L$^{-1}$, stormflow median = 2.50 mg N L$^{-1}$; p < 0.005, Figure S3). Both
GWN and GUN had similar NO$_3^-{}_{Atm}$ concentrations between baseflow and storm events (GWN
baseflow median = 0.05 mg N L$^{-1}$, stormflow median = 0.06 mg N L$^{-1}$, p > 0.05, GUN baseflow
median = 0.04 mg N L$^{-1}$, stormflow median = 0.06 mg N L$^{-1}$, p > 0.05, Figure S3).

Similar to NO$_3^-$ concentrations and isotopes, $\delta^{18}$O-H$_2$O values exhibited greater variability

between baseflow and peak streamflow in GWN than in GUN. From baseflow to approximately peak
streamflow, $\delta^{18}$O-H$_2$O shifted by an absolute average of 2.1 ‰ at GWN but only 0.6 ‰ at GUN (Table
S2). These shifts correspond to an average event-water fraction at peak storm discharge of 0.75 ±0.13
at GWN and 0.27 ±0.23 at GUN (Table S2). Event-water fraction uncertainty was relatively large for
several events due to small separation between $\delta^{18}$O-H$_2$O end members. For example, rainfall and pre-
event baseflow end members were separated by only 0.5 ‰ during the 7/22/19 event at GUN, resulting
in uncertainty of event-water fractions exceeding 1 (Tables S1 and S2).

Storms events have an outsized impact, relative to baseflow, on NO$_3^-{}_{Atm}$ export at GWN, as

indicated by $DF > 1$ for 7 of 8 sampled events (mean = 2.6 ±0.4; Figure 2). The opposite relationship
was observed for NO$_3^-{}_{Terr}$ at GWN ($DF \leq 1$ for all sampled events, mean = 0.5 ±0.1) indicating that
baseflow has an outsized impact on NO$_3^-{}_{Terr}$ loads relative to storm events. Conversely, $DF$ values at
GUN were approximately 1 for both NO$_3^-{}_{Atm}$ (mean = 1.1 ±0.2) and NO$_3^-{}_{Terr}$ (mean = 1.0 ±0.1),
indicating that neither baseflow nor stormflow disproportionately impacted stream NO$_3^-$ loads (Figure





2). Event-water fractions were positively, though not significantly, related to $DF$ of $NO_3^-{}_{Atm}$ ($\tau = 0.32$,
p = 0.09) and negatively related to $DF$ of $NO_3^-{}_{Terr}$ across both watersheds (Figure 4; $\tau = -0.32$, p <
0.05). In GWN, the total rainfall depth for a given event was positively correlated with the fraction of
deposited $NO_3^-$ that was exported in streamwater during the same event ($\tau = 0.74$, p < 0.05), but there
was no relationship for GUN (Figure 5). Additionally, there was a 1:1 relationship between the event
$NO_3^-{}_{Atm}$ deposition on impervious surfaces and the event $NO_3^-{}_{Atm}$ streamwater export at GWN ($r^2$ =
0.55, p < 0.05), but not at GUN (Figure 6). $NO_3^-{}_{Atm}$ load estimates using traditional methods
(concentration from a single grab sample multiplied by mean daily discharge) were biased (range = -
197 % – 123 %, median absolute value = 36 %) relative to $NO_3^-{}_{Atm}$ load estimates using the multiple
samples we collected across the storm hydrograph for the two events that encompassed a full day.
**4 Discussion**

Hydrologic effects of impervious surfaces likely drive the disproportionate impact of storm

events on $NO_3^-{}_{Atm}$, and of baseflow on $NO_3^-{}_{Terr}$, in the more developed watershed (GWN). Impervious
surfaces increase peak storm runoff (Arnold and Gibbons, 1996; Walsh et al., 2005), but differences in
peak discharge alone are not the sole explanation for the contrasting results of $DF$ for $NO_3^-{}_{Terr}$ and
$NO_3^-{}_{Atm}$ between the watersheds. Sampled events with overlapping $f_{Runoff}$ between sites (i.e., similar x-
axis values on Figure 2) indicate that the difference between $f_{NO3}$ for $NO_3^-{}_{Terr}$ and $NO_3^-{}_{Atm}$ is much
greater at the more developed (GWN) than the less developed watershed (GUN; i.e., different y-axis values
on Figure 2). Thus, it is the overland routing of rainfall, and $NO_3^-{}_{Atm}$ dissolved therein, that likely
contributes to the outsized impact of storm events on $NO_3^-{}_{Atm}$ in the developed watershed. Although
both watersheds show a positive relationship between event-water fractions and $DF$ of $NO_3^-{}_{Atm}$ (p =
0.09, Figure 4), event-water fractions are much greater in the more developed watershed, GWN (green
triangles in Figure 4). Higher event-water fractions promote greater export of $NO_3^-{}_{Atm}$ by reducing the
potential for biological processing or retention. Our results provide evidence (i.e., increased event-
water fractions, proportional streamwater export of impervious $NO_3^-{}_{Atm}$ deposition) for the mechanism
(i.e., direct routing of rainfall $NO_3^-{}_{Atm}$ to streams) that generates increased $NO_3^-{}_{Atm}$ export in more
developed watersheds, which thus expands on previous research demonstrating that more developed



watersheds export relatively more $NO_3^-{}_{Atm}$ (Buda and DeWalle, 2009; Burns et al., 2009; Kaushal et
al., 2011; Bostic et al., 2021).

Our study collected samples across the storm hydrograph and measured $\Delta^{17}O$ of $NO_3^-$, which

provided a more accurate load estimates of, and insights into, storm $NO_3^-{}_{Atm}$ export than $\delta^{18}O$ of
$NO_3^-$. For example, estimates of daily $NO_3^-{}_{Atm}$ loads were biased by a median absolute value of 36%
using standard methods (i.e., daily average discharge multiplied by $NO_3^-{}_{Atm}$ concentration, estimated
using $\Delta^{17}O$, of a single grab sample; Tsunogai et al., 2014; Rose et al., 2015b; Nakagawa et al., 2018)
when compared to "true" daily loads calculated using samples collected across the storm hydrograph
from two events that encompassed a full day. Additionally, use of $\Delta^{17}O$ generally provides more
certain estimates of $NO_3^-{}_{Atm}$ fractions and concentrations than $\delta^{18}O$ because biological processing
(e.g., assimilation, denitrification) can change the $\delta^{18}O$ of $NO_3^-$ and generate large uncertainty (±
~30‰, Kendall et al., 2007) in the $\delta^{18}O$-$NO_3^-{}_{Terr}$ end-member and ultimately estimates of $NO_3^-{}_{Atm}$
(Tsunogai et al., 2016). $\Delta^{17}O$ of $NO_3^-$, due to its mass-independent fractionation origin, is not subject
to the same variability associated with biological processing as $\delta^{18}O$, thereby decreasing uncertainty in
$NO_3^-{}_{Atm}$ estimates (Young et al., 2002; Michalski et al., 2004; Kendall et al., 2007). Indeed, average
event $NO_3^-{}_{Atm}$ fractions (i.e., $\frac{NO_3^-{}_{Atm}}{NO_3^-{}_{Total}}$) would have been underestimated by an average of 3% (range = 0
– 7 %) at both sites if using $\delta^{18}O$-$NO_3^-$ only (Figure S4), but with a greater effect at the more
developed site (GWN). An average underestimate of 3% may appear minor, but it is notable
considering that event $NO_3^-{}_{Atm}$ fractions averaged 2% and 10% in the less and more developed
watersheds, respectively. Increased accuracy of $NO_3^-{}_{Atm}$ export during storm events combined with the
*DF* conceptual framework (Figure 2) provides a relatively simple means of assessing whether storm
events or baseflow have an outsized impact on $NO_3^-$ source export. More accurate estimates of
$NO_3^-{}_{Atm}$ export also allow for more quantitative investigations into the role of impervious surfaces in
routing event rainfall $NO_3^-{}_{Atm}$ to streams.

Impervious areas in the developed watershed are effective conduits of $NO_3^-{}_{Atm}$ to surface

waters, as demonstrated by the approximately proportional relationship between event streamwater
$NO_3^-{}_{Atm}$ export and event $NO_3^-{}_{Atm}$ deposition on impervious surfaces (Figure 6). This relationship



provides evidence, in addition to higher event-water fractions (Figure 4), for the mechanism of
impervious surfaces enhancing export of $NO_3^-{}_{Atm}$ during storm events. The 1:1 correspondence of this
relationship is surprising, however. For 100% of rainfall $NO_3^-{}_{Atm}$ on impervious surfaces to be
exported as streamwater during a given event (i.e., 1:1 relationship), all impervious area in the
watershed would have to be hydrologically connected to surface waters (i.e., effective impervious
areas; Shuster et al., 2005). In a mesoscale (84 km$^2$) and heterogeneous watershed such as GWN, the
total impervious area is not equivalent to effective impervious area. Rather, many impervious surfaces
drain onto pervious surfaces, or are "ineffective" at directly routing precipitation to channels (Walesh,
1989; but we note that certain pervious surfaces, such as reclaimed mine lands, effectively function as
impervious, e.g., Negley and Eshleman 2006). It is likely that the observed 1:1 relationship (Figure 6)
is additionally affected by flushing of dry $NO_3^-{}_{Atm}$ deposition from effective impervious areas. Dry
$NO_3^-$ deposition, similar to wet deposition, inherits positive $\Delta^{17}O$ values (~15 – 30 ‰; Nelson et al.,
2018) and is generally higher in urban relative to rural areas both locally (Lovett et al., 2000; Bettez
and Groffman, 2013) and globally (Decina et al., 2019). Thus, flushing of dry $NO_3^-$ deposition residing
on impervious surfaces (or on surfaces such as leaves that can wash onto impervious surfaces) during
storm events could contribute to the 1:1 relationship observed in the more developed watershed (green
circles in Figure 6).

$\Delta^{17}O$ of $NO_3^-$ can additionally be used to "correct" $\delta^{15}N$ and $\delta^{18}O$ values (eqs. 7 and 8) to

better indicate isotope values of terrestrial $NO_3^-$ sources (Dejwakh et al., 2012). Values of both
$\delta^{15}N_{Terr}$ and $\delta^{18}O\text{-}NO_3^-{}_{Terr}$ during storm events fall within the range of values that are typical of natural
"soil" and fertilizer (Kendall et al., 2007), but interestingly, $NO_3^-{}_{Terr}$ isotope values decreased during
storm events relative to baseflow in both watersheds (though not significantly for $\delta^{15}N$ in GUN; Figure
3). This shift to lower $\delta^{15}N_{Terr}$ and $\delta^{18}O\text{-}NO_3^-{}_{Terr}$ values during storm events may reflect the flushing of
less "processed" $NO_3^-$ sources from upper soil horizons (Creed et al., 1996), as processing (e.g.,
denitrification) generally leaves the remaining $NO_3^-$ with more positive $\delta^{15}N$ and $\delta^{18}O$ values due to
biologically-mediated fractionation (Denk et al., 2017). Impervious surfaces in the developed
watershed likely reduce flushing of this lower $\delta^{18}O\text{-}NO_3^-{}_{Terr}$ by restricting infiltration, but 30% of this
watershed is not "developed" (and a higher percentage contains pervious surfaces), which likely



contributes to the similarity in $NO_3^-{}_{Terr}$ isotope patterns between study watersheds. Relatively lower
$\delta^{18}O$- $NO_3^-{}_{Terr}$ values during storm events relative to baseflow, and associated insights into watershed-
scale N biogeochemistry, were only realized by using $\Delta^{17}O$ to "correct" $\delta^{18}O$ values. Without this
correction, $\delta^{18}O$-$NO_3^-$ during storm events is strongly influenced by elevated $\delta^{18}O$ of $NO_3^-{}_{Atm}$, as
shown by the similar patterns between $\Delta^{17}O$ and "uncorrected" $\delta^{18}O$ in the more developed watershed
(Figure 3).

Large inputs and stores of N associated with agricultural activity likely contribute to baseflow

and storm events having similar impacts on $NO_3^-{}_{Terr}$ and $NO_3^-{}_{Atm}$ export in the mixed
agricultural/forested watershed (GUN). $DF$s of both $NO_3^-{}_{Terr}$ and $NO_3^-{}_{Atm}$ were approximately 1,
indicating that loads are primarily explained by changes in discharge. Nutrients, including $NO_3^-$,
showing similar patterns (loads explained primarily by discharge) over annual time-scales have been
attributed to large stores of $NO_3^-$ associated with agricultural inputs (Basu et al., 2010; Thompson et
al., 2011). With significant agricultural land-use, both currently (41.3% in 2016; Table 1) and
historically (~58% in 1960; O'Bryan and McAvoy, 1966), and consistently high $NO_3^-$ concentrations
in streamwater, GUN likely has large stores of $NO_3^-$ in soil and groundwater. Interestingly, our results
demonstrate the control of discharge on $NO_3^-{}_{Terr}$ and $NO_3^-{}_{Atm}$ loads over storm-event time scales,
suggesting that large reservoirs of $NO_3^-$ contribute to streamwater export of nutrients across varied
flow conditions and not just baseflow.

The combination of our results with projections of increasing frequency of intense

precipitation events (Najjar et al., 2010; Walsh et al., 2014) and increasing urban and suburban sprawl
(Jantz et al., 2005; Seto et al., 2012) suggest that $NO_3^-{}_{Atm}$ may become a relatively more important
$NO_3^-$ source to downstream waters, assuming no change in $NO_3^-$ deposition rates. This assumption
may not be valid everywhere, however; for example, $NO_3^-$ deposition is declining locally (i.e., mid-
Atlantic USA; Li et al., 2016) but increasing across many regions (i.e., east Asia; Liu et al., 2013). In
our more developed watershed, the positive correlation between rainfall and the fraction of deposited
$NO_3^-$ exported in streamwater (Figure 5) suggests that large storm events may export proportionally
greater fractions of rainfall $NO_3^-{}_{Atm}$ in urbanizing catchments and increased loads of $NO_3^-{}_{Atm}$ to
downstream waters. Best management practices in developed watersheds (e.g., stormwater control



measures) can mitigate these potential impacts by increasing infiltration of rainfall (and $NO_3^-$
dissolved in rainfall) and reducing hydrologic connectivity of overland flowpaths (i.e., decrease
effective impervious areas; Lee and Heaney, 2003; Walsh et al., 2009), both of which may reduce the
load of $NO_3^-{}_{Atm}$ and the proportion of "event" water in streams during storm events. Such practices
may additionally reduce $NO_3^-{}_{Terr}$ loads by stimulating denitrification (Bettez and Groffman, 2012), but
could also increase the importance of baseflow in $NO_3^-$ export due to increased infiltration. Thus,
monitoring of both baseflow and storm events is necessary to quantify potential changes and make
targeted water-quality management decisions. Finally, best management practices intended to reduce
$NO_3^-{}_{Atm}$ loads in developed watersheds via increased infiltration may provide numerous co-benefits,
including reduced runoff (Hood et al., 2007) and higher baseflow (Fletcher et al., 2013), both of which
could help restore aquatic ecosystems impacted by urbanization (Walsh et al., 2005).
**5. Conclusion**
We found that stormflow has a disproportionately large impact on $NO_3^-{}_{Atm}$ export whereas
baseflow has a disproportionately small impact on $NO_3^-{}_{Terr}$ export in a moderately developed
watershed. In contrast, neither stormflow nor baseflow have an outsized impact on $NO_3^-{}_{Atm}$ or
$NO_3^-{}_{Terr}$ export in a mixed land-use watershed with significant agriculture. Hydrologic connectivity of
overland flow paths associated with impervious surfaces likely promote rapid transport of $NO_3^-{}_{Atm}$ to
streams during storm events in the more developed watershed, with higher rainfall storms exporting a
greater fraction of deposited $NO_3^-$ than lower rainfall events and event $NO_3^-{}_{Atm}$ streamwater export
approximately equaling rainfall $NO_3^-{}_{Atm}$ on impervious surfaces. Large reserves of new and/or legacy
agricultural-associated nitrogen in soils in the mixed land-use watershed likely influenced the similar
response of $NO_3^-{}_{Atm}$ or $NO_3^-{}_{Terr}$ to stormflow and baseflow.
**Appendices**
Not applicable.



**Code availability**

Not applicable.

**Data availability**

Available upon request.

**Author contributions**

DMN and KNE: Conceptualization, Methodology, Writing – Review and Editing, Supervision, Funding Acquisition

JTB: Conceptualization, Methodology, Investigation, Formal Analysis, Writing – Original Draft, Writing – Review and Editing, Visualization, Funding Acquisition
**Acknowledgements**

Thanks to Pavithra Pitumpe Arachchige and Jim Garlitz for $NO_3^-$ concentration analysis. Robert Hirsch of the U.S. Geological Survey provided guidance on WRTDS-K and R scripts for estimating $NO_{3-Total}$ uncertainty. DMN, KNE, and JTB received support from Maryland Sea Grant under award NA14OAR4170090 R/WS-3 from the National Oceanic and Atmospheric Administration, U.S. Department of Commerce. JTB received support from Maryland Sea Grant under award SA75281900-A from the National Oceanic and Atmospheric Administration, U.S. Department of Commerce. This material is based upon work supported by the National Science Foundation Graduate Research Fellowship (to JTB) under Grant No. 1840380. Any opinion, findings, conclusions, recommendations expressed in this material are those of the authors and do not necessarily reflect the views of the National Science Foundation.

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





**Tables**

**Table 1. Watershed attributes.**

| Watershed | Area (ha) | Land-Use (%) | | | | MAT (°C) | MAP (cm) | Lithology (%) | | |
|---|---|---|---|---|---|---|---|---|---|---|
| | | Forest | Agriculture | Developed | Impervious | | | Un-consolidated | Crystalline | Carbonate |
| Gwynns Falls (GWN) | 8400 | 23.4 | 5.0 | 70.1 | 14.6 | 12.7 | 113.5 | 0 | 95.1 | 4.9 |
| Gunpowder Falls (GUN) | 41400 | 45.4 | 41.3 | 10.9 | 0.3 | 11.9 | 116.0 | 0 | 99.8 | 0.2 |

**Land-use percentages were calculated from the 2016 National Land Cover Database, impervious is the sum of medium and high intensity developed land-use classes; agricultural land represents the sum of both cultivated crop and pasture/hay land classes (Homer et al., 2020). Land use percentages do not sum to 100% as all land use classes are not listed (e.g. open water, wetlands). MAT = Mean Annual Temperature, MAP = Mean Annual Precipitation. Note that MAT and MAP cover the time period from 1981-2010 (PRISM, 2014). Lithology data were obtained from Zhang et al. 2019.**





**Figures**

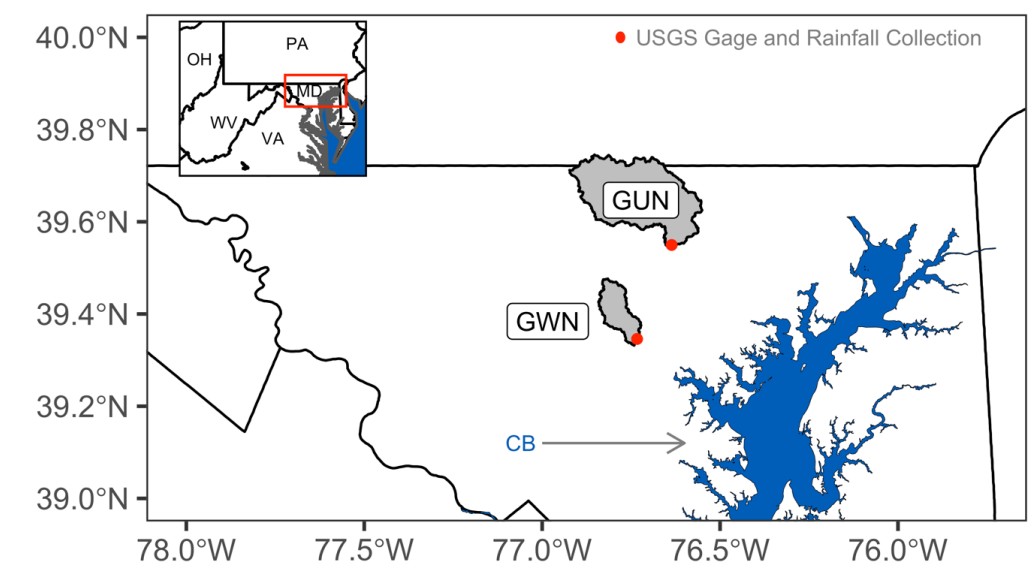

Figure 1. Site map showing watershed boundaries (GWN = Gwynns Falls, GUN = Gunpowder Falls), United States Geology
Survey (USGS) gaging stations and rainfall collection sites, and Chesapeake Bay (CB) location. Inset map shows relative position
5   of watersheds in Maryland (MD) relative to neighboring states (PA = Pennsylvania, OH = Ohio, WV = West Virginia, VA =
Virginia).





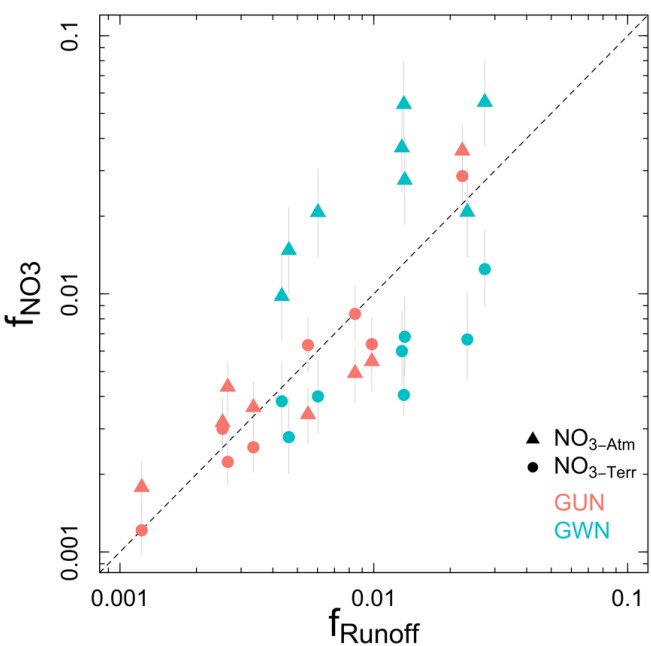

**Figure 2. Fraction of NO$_3^-$ loads (fNO3; separated by NO$_3^-$Terr, circles, and NO$_3^-$Atm, triangles) and discharge (fRunoff) during the study duration (14 months) represented by sampled storm events (n = 8). Points falling above the dashed line (1:1 line) indicate**
10 **storm events have an outsized impact on NO$_3^-$ loads and points falling below the line indicate baseflow has an outsized impact on NO$_3^-$ loads. Points on or near the 1:1 line indicate a chemostatic response, in which storms nor baseflow have an outsized impact on NO$_3^-$ loads.**



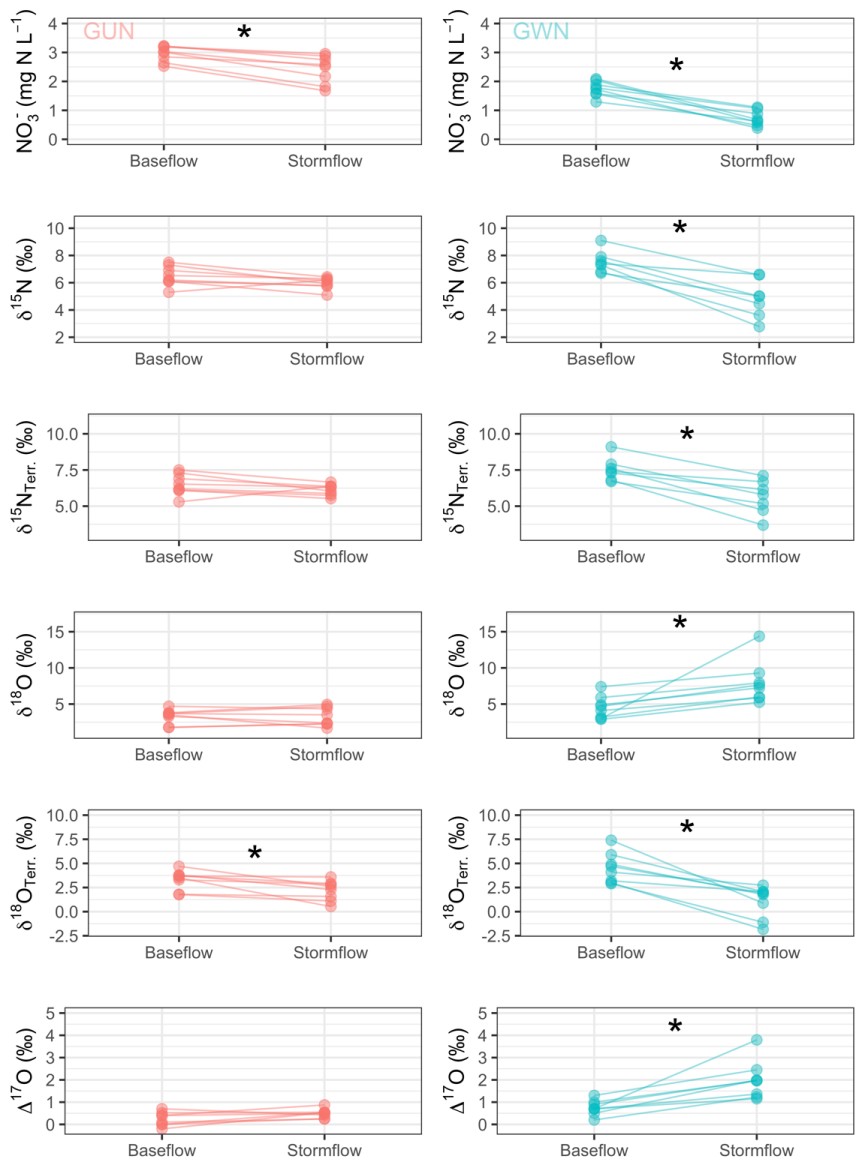

**Figure 3. Event mean $NO_3^-$ concentrations and $\delta^{15}N$, $\delta^{15}N_{Terr}$, $\delta_{18}O$, $\delta^{18}O_{Terr}$, and $\Delta^{17}O$ values of $NO_3^-$ for samples collected during storm events paired with the corresponding baseflow sample preceding the event. Asterisk (*) indicates significant difference at p < 0.05 as determined using a Wilcoxon ranked-sum test.**



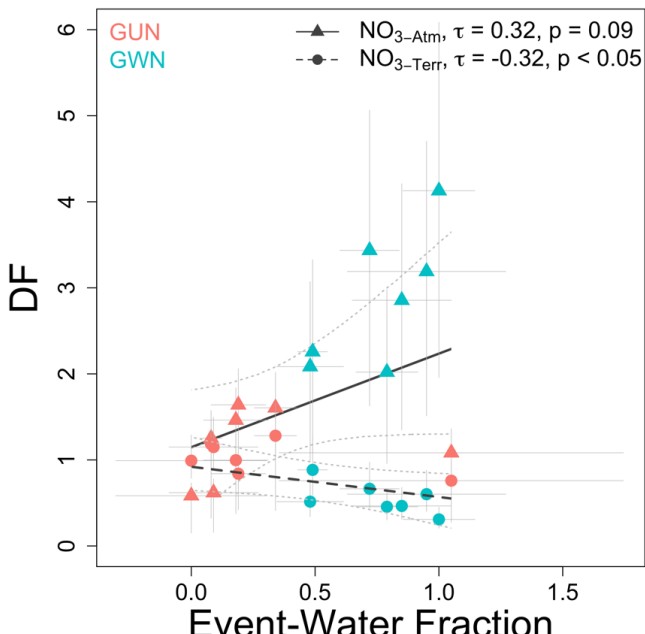

**Figure 4. Disproportionality factor (DF) and event-water fraction for $NO_{3\ Atm}^{-}$ (triangles) and $NO_{3\ Terr}^{-}$ (circles). Event-water fraction and DF are positively, but not significantly correlated for $NO_{3\ Atm}^{-}$ ($\tau = 0.32$, $p = 0.09$) while event-water fraction and DF are significantly, negative correlated for $NO_{3\ Terr}^{-}$ ($\tau = -0.32$, $p < 0.05$) across both watersheds. The thin, dotted line shows bootstrapped 95% confidence intervals.**



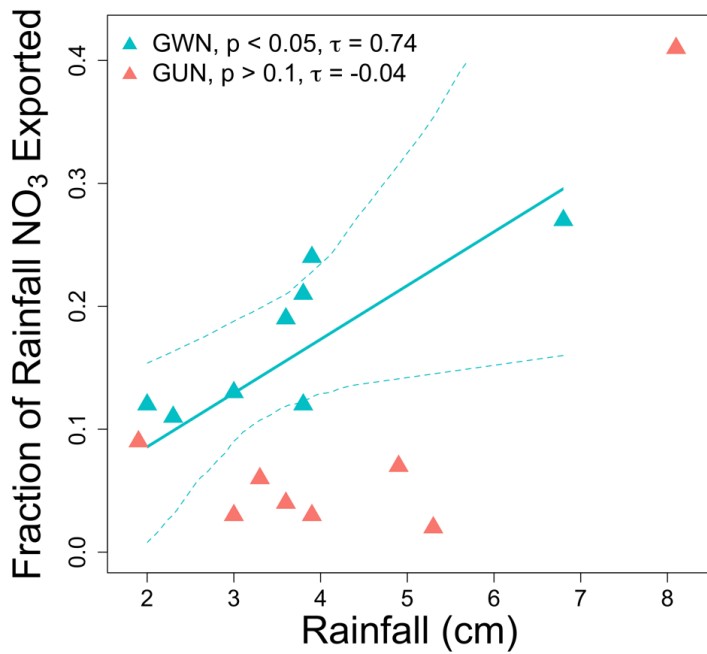

**Figure 5. The fraction of NO$_3^-$ in rainfall that is exported in streamwater during the same event is positively significantly related with total event rainfall at GWN (p < 0.05, τ = 0.74) but not at GUN (p > 0.1, τ = -0.04). The solid line is the Theil-Sen slope and the thin, dotted line shows the bootstrapped 95% confidence intervals.**



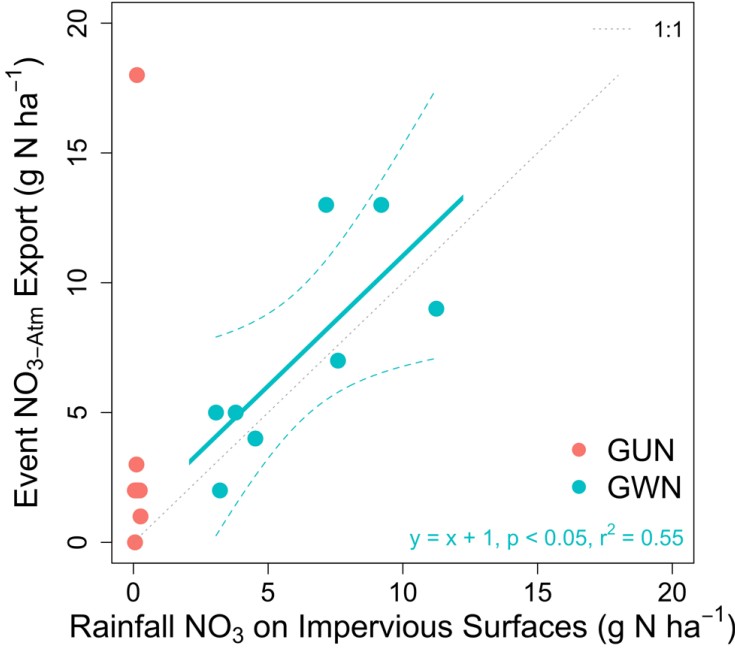

**Figure 6.** The event $NO_3^-{}_{Atm}$ yield (in g N ha$^{-1}$) has a 1:1 relationship with the estimated rainfall $NO_3^-{}_{Atm}$ deposition on impervious surfaces (in g N ha$^{-1}$) at GWN (slope = 1.00, intercept = 1, $r^2$ = 0.55, $p < 0.05$), but not at GUN.