# Peer review of "Downpour Dynamics: Outsized impacts of storm events"

_Biogeosciences, 2023_

## Referee Comment (RC1)

By estimating the concentration of stream nitrate ($NO_3^-$), isotopic compositions of $NO_3^-$ ($\delta^{15}N$, $\delta^{18}O$, and $\Delta^{17}O$), and water flux during base flow and storm events, this manuscript investigated the exportation of $NO_3^-{}_{terr}$ and $NO_3^-{}_{atm}$ in two different watersheds, where two watersheds have different land-use. Compared to the mixed agricultural/forested watershed, the developed urban watershed exported more $NO_3^-{}_{atm}$ during storm events, which was explained by impervious surfaces that hydrologically connect runoff to channels to facilitate the export of $NO_3^-{}_{atm}$ during storm events. In addition, the disproportionality factor was proposed to quantify the disproportionate effect of $NO_3^-{}_{terr}$ and $NO_3^-{}_{atm}$ compared to the runoff during storm events.

While the paper is nicely written, I have one major concern about the $\Delta^{17}O$ of atmospheric $NO_3^-$ in rainfall. In the author's past study, they reported the mean value of $\Delta^{17}O$ of atmospheric $NO_3^-$ in three nearly stations was +25.1 ‰ (Bostic et al., 2021; figures of following), which the value was in accordance with other past studies (e.g., +26.3 ± 3 ‰; Tsunogai et al., 2016, +26.1 ± 3.5 ‰; Hale et al., 2014, and +20 ~ +30 ‰; Michalski et al., 2003) in the similar latitudes. On the other hand, the mean value of $\Delta^{17}O$ of atmospheric $NO_3^-$ in this study was +20.2 ± 2.8‰ (Table S1), which the value was seems significantly smaller than the past studies. The concern should be resolved before publication.

[Figure]

Specific comments:

Line 140-147: The calculated $NO_3^-{}_{atm}$ Deposition should compare with similar past studies to verify the accuracy of the data.

Line 240: Is DF=1 here?

L370-373: The interpretation of low $\delta^{15}N_{Terr}$ and $\delta^{18}O_{Terr}$ during storm events was reasonable; the same phenomenon has also been reported by a recent study (Ding et al., 2022). However, there is another possibility, the shorter residence time of stream $NO_3^-$ during storm events could cause smaller biologically-mediated fractionation (having not enough time for bioreactions of fractionation) than normal time; thus, the exported $NO_3^-$ showed low values of $\delta^{15}N_{Terr}$ and $\delta^{18}O_{Terr}$, rather than the addition of new $NO_3^-$. In addition, the reason why the $\delta^{15}N_{Terr}$ in GUN watershed didn't show low values (and the weaker significance of $\delta^{18}O_{Terr}$) also should be discussed in the manuscript. Because GUN watershed showed higher land-use of forest and agriculture (Table 1), the flushing effect should be stronger in GUN watershed.

Figure 4: While other figures had 8 points, there were only 7 points in Figure 4, Figure S4 also.

Figure 5: The calculation of the fraction of rainfall $NO_3^-$ exported (y-axis) should be expressed in section 2 of the manuscript.
How about using the intensity of rainfall (unit: cm/h) as the x-axis?

Figure S2: It seems many stream water samples were sampled during baseflow and storm periods. Did the authors analyze the isotopic compositions ($\delta^{15}N$, $\delta^{18}O$, and $\Delta^{17}O$ of $NO_3^-$) of all these samples? If the authors did, they can list these data in supplementary and the number of analyzed in the manuscript, not only the mean value (Figure 3; Table S3).

**Reference**

Bostic, J. T., Nelson, D. M., Sabo, R. D., & Eshleman, K. N. (2021). Terrestrial Nitrogen Inputs Affect the Export of Unprocessed Atmospheric Nitrate to Surface Waters: Insights from Triple Oxygen Isotopes of Nitrate. *Ecosystems*. https://doi.org/10.1007/s10021-021-00722-9

Ding, W., Tsunogai, U., Nakagawa, F., Sambuichi, T., Sase, H., Morohashi, M., and Yotsuyanagi, H.: Tracing the source of nitrate in a forested stream showing elevated concentrations during storm events, Biogeosciences, 19, 3247–3261, https://doi.org/10.5194/bg-19-3247-2022, 2022.

Hale, R. L., Turnbull, L., Earl, S., Grimm, N., Riha, K., Michalski, G., et al. (2014). Sources and transport of nitrogen in arid urban watersheds. *Environmental Science and Technology*, *48*(11), 6211–6219. https://doi.org/10.1021/es501039t

Michalski, G., Scott, Z., Kabiling, M., & Thiemens, M. H. (2003). First measurements and modeling of $\Delta 17O$ in atmospheric nitrate. *Geophysical Research Letters*, *30*(16), 3–6. https://doi.org/10.1029/2003GL017015

Tsunogai, U., Miyauchi, T., Ohyama, T., Komatsu, D. D., Nakagawa, F., Obata, Y., et al. (2016). Accurate and precise quantification of atmospheric nitrate in streams draining land of various uses by using triple oxygen isotopes as tracers. *Biogeosciences*, *13*(11), 3441–3459. https://doi.org/10.5194/bg-13-3441-2016

---

## Author Comment (AC1)

*Author replies are indented and italicized.*

**Reviewer 1:**

By estimating the concentration of stream nitrate (NO3-), isotopic compositions of NO3 - (δ15N, δ18O, and Δ17O), and water flux during base flow and storm events, this manuscript investigated the exportation of NO3 - terr and NO3 - atm in two different watersheds, where two watersheds have different land-use. Compared to the mixed agricultural/forested watershed, the developed urban watershed exported more NO3 - atm during storm events, which was explained by impervious surfaces that hydrologically connect runoff to channels to facilitate the export of NO3 - atm during storm events. In addition, the disproportionality factor was proposed to quantify the disproportionate effect of NO3 - terr and NO3 - atm compared to the runoff during storm events.

While the paper is nicely written, I have one major concern about the Δ17O of atmospheric NO3 - in rainfall. In the author's past study, they reported the mean value of Δ17O of atmospheric NO3 - in three nearly stations was +25.1 ‰ (Bostic et al., 2021; figures of following), which the value was in accordance with other past studies (e.g., +26.3 ± 3 ‰; Tsunogai et al., 2016, +26.1 ± 3.5 ‰; Hale et al., 2014, and +20 ~ +30 ‰; Michalski et al., 2003) in the similar latitudes. On the other hand, the mean value of Δ17O of atmospheric NO3 - in this study was +20.2 ± 2.8‰ (Table S1), which the value was seems significantly smaller than the past studies. The concern should be resolved before publication.

*Thank you for your comments regarding D17O of our rainfall samples. The reviewer is correct that the average D17O value of NO3-Atm in rainfall ( in this manuscript 20.2 per mil) is lower than that in Bostic et al., 2021 (25.2 per mil). However, both values are within measured ranges reported by studies in similar latitudes (Michalski et al., 2003; Xia et al., 2019). There is no reason to believe the different values are related to methodological/analytical issues, as the samples in both studies were analyzed in the same lab using the same methodology, and instrumental precision and accuracy of D17O-NO3 data was similar between the studies. We suspect the lower values in this manuscript relative to Bostic et al., 2021 are due to one of two most likely factors:*

> *(1) Samples were not collected in the same location. In Bostic et al., 2021, precipitation samples (weekly composites) were collected from three National Atmospheric Deposition Program sites from October 2016-September 2017. The locations of those sites are shown as red triangles in the figure below from Bostic et al., 2021. In the present manuscript, precipitation samples were collected (during precipitation events) at the outlet of the two watersheds (GUN and GWN on the below figure) from September 2018-October 2019. Importantly, GWN is a highly urbanized watershed. Previous work has shown that oxidation pathways of NO and NO2 can differ between urban and rural areas, resulting in lower D17O-NO3 values in urban deposition (Li et al., 2022; Nelson et al., 2018). The different sampling frequencies between studies (weekly vs event-based) could*

*also potentially influence the observed differences in NO3-D17O values.*

[Figure]

(2) *Relatively few rainfall samples were collected during winter in the present study. Previous studies (list here; Xia et al., 2019; Nelson et al., 2018; Huang et al., 2020), have shown a clear seasonal pattern of D17O of NO3-Atm, with higher values in winter and lower values in summer. Six of the eight storm events sampled in the present manuscript occurred between the months of May - October. Included below is a figure of D17O of NO3-Atm for all sampled events in this manuscript, along with the samples collected from Bostic et al., 2021. Samples from the present manuscript approximately follow the seasonal pattern that was observed in Bostic et al., 2021. The combined effects of relatively more samples collected in summer than winter and differing atmospheric chemistry of urban areas likely contributed to the slightly lower average of D17O of rainfall NO3 in this study compared to our previous research in the region (i.e., Bostic et al., 2021).*

[Figure]

Specific comments:

Line 140-147: The calculated NO3- atm Deposition should compare with similar past studies to verify the accuracy of the data.

> *We used the same procedure as previous studies (Lovett et al., 2000; Nelson et al., 2018; Huang et al., 2020) to estimate NO3 deposition (equation 1 in the manuscript).*

Line 240: Is DF=1 here?

> *Thank you for catching this incomplete sentence. This sentence has been revised to:*

> > *For example, an event with DF = 4 indicates that a given storm exported 4× more $NO_3^-$ than water whereas an event with DF = 0.5 indicates that a storm exported 2× less $NO_3^-$ than water, after both have been normalized to annual amounts.*

L370-373: The interpretation of low δ15NTerr and δ18OTerr during storm events was reasonable; the same phenomenon has also been reported by a recent study (Ding et al., 2022). However, there is another possibility, the shorter residence time of stream NO3 - during storm events could cause smaller biologically-mediated fractionation (having not enough time for bioreactions of fractionation) than normal time; thus, the exported NO3 - showed low values of δ15NTerr and δ18OTerr, rather than the addition of new NO3 -. In addition, the reason why the δ15NTerr in GUN watershed didn't show low values (and the weaker significance of δ18OTerr) also should be discussed in the manuscript. Because GUN watershed showed higher land-use of forest and agriculture (Table 1), the flushing effect should be stronger in GUN watershed.

> *Thank you for your comment regarding interpretation of δ15NTerr and δ18OTerr and your suggestion regarding the possible influence of shorter residence time of stream NO3 during storm events. We agree that reduced residence time could play a role and have added a sentence to the discussion. We have also added two new sentences further elaborating on the differences between GUN and GWN. The revised paragraph is below with additions highlighted:*

> > *$D^{17}O$ of $NO_3^-$ can additionally be used to "correct" $d^{15}N$ and $d^{18}O$ values (eqs. 7 and 8) to better indicate isotope values of terrestrial $NO_3^-$ sources (Dejwakh et al., 2012). Values of both $d^{15}N_{Terr}$ and $d^{18}O\text{-}NO_3^-_{Terr}$ during storm events fall within the range of values that are typical of natural "soil" and fertilizer (Kendall et al., 2007), but interestingly, $NO_3^-_{Terr}$ isotope values decreased during storm events relative to baseflow in both watersheds (though not significantly for $d^{15}N$ in GUN; Figure 3). This shift to lower $d^{15}N_{Terr}$ and $d^{18}O\text{-}NO_3^-_{Terr}$ values during storm events may reflect the flushing of less "processed" $NO_3^-$ sources from upper soil*

*horizons (Creed et al., 1996), as processing (e.g., denitrification) generally leaves the remaining $NO_3^-$ with more positive $d^{15}N$ and $d^{18}O$ values due to biologically-mediated fractionation (Denk et al., 2017).* ==*Lower $d^{15}N_{Terr}$ during storm events relative to baseflow was not statistically significant in the mixed agricultural/forested watershed (GUN), but this was due to a single event in which $d^{15}N_{Terr}$ increased from baseflow to stormflow.*== *Impervious surfaces in the developed watershed likely reduce flushing of this lower $d^{18}O$-$NO_3^-_{Terr}$ by restricting infiltration, but 30% of this watershed is not "developed" (and a higher percentage contains pervious surfaces), which likely contributes to the similarity in $NO_3^-_{Terr}$ isotope patterns between study watersheds.* ==*Additionally, relatively lower $NO_3^-_{Terr}$ isotope values in storm events could be due to reduced in-stream $NO_3^-$ uptake (e.g., assimilation, denitrification) during periods of elevated discharge (Grimm et al., 2005). Biological $NO_3^-$ uptake generally fractionates against heavier isotopes which increases isotope ratios of the remaining $NO_3^-$ (Kendall et al., 2007). If in-stream $NO_3^-$ uptake rates are reduced during high flows, the resulting effect could contribute to the lower $NO_3^-_{Terr}$ isotope values during storm events.*== *Relatively lower $d^{18}O$- $NO_3^-_{Terr}$ values during storm events relative to baseflow, and associated insights into watershed-scale N biogeochemistry, were only realized by using $D^{17}O$ to "correct" $d^{18}O$ values. Without this correction, $d^{18}O$-$NO_3^-$ during storm events is strongly influenced by elevated $d^{18}O$ of $NO_3^-_{Atm}$, as shown by the similar patterns between $D^{17}O$ and "uncorrected" $d^{18}O$ in the more developed watershed (Figure 3).*

Figure 4: While other figures had 8 points, there were only 7 points in Figure 4, Figure S4 also.

> *Thanks for catching this mistake. A pre-storm, baseflow sample was not collected for the first event in either watershed. All figures comparing baseflow to stormflow dynamics or figures that require a baseflow sample (e.g., event water fraction) should only have 7 data points. We have added the following text to the methods for clarification:*
>
> > *A pre-event baseflow sample was not collected for the first storm, thus any figures or analyses that compare pre-event baseflow to event mean concentrations or event-water fractions have seven data points.*
>
> *We note that this does not change the statistical significance for any results.*

Figure 5: The calculation of the fraction of rainfall NO3 - exported (y-axis) should be expressed in section 2 of the manuscript. How about using the intensity of rainfall (unit: cm/h) as the x-axis?

> *We have added the following text and equation to the methods section:*
>
> > *The fraction of rainfall $NO_3^-$ exported on an event basis was calculated as:*

$$\text{Fraction of rainfall } NO3^- \text{ exported} = \frac{NO_{3-Atm}^- Yield \ (g \ N \ ha^{-1})}{NO_{3-Atm}^- Deposition \ (g \ N \ ha^{-1})} \ \text{(eq. 7)}$$

*where event $NO_{3\ Atm}^-$ deposition was calculated using eq. 1 and event $NO_{3\ Atm}^-$ yield was calculated using eq. 5. We appreciate the suggestion to use intensity of rainfall on the x-axis, but unfortunately we lack such data.*

Figure S2: It seems many stream water samples were sampled during baseflow and storm periods. Did the authors analyze the isotopic compositions ($\delta 15N$, $\delta 18O$, and $\Delta 17O$ of NO3 - ) of all these samples? If the authors did, they can list these data in supplementary and the number of analyzed in the manuscript, not only the mean value (Figure 3; Table S3).

*We did analyze the isotopic composition of all the samples shown in Figure S2. We will add a supplementary table with this data.*

**References**

Bostic JT, Nelson DM, Sabo RD, Eshleman KN. 2021. Terrestrial nitrogen inputs affect the export of unprocessed atmospheric nitrate to surface waters: Insights from triple oxygen isotopes of nitrate. Ecosystems. 25, 1384–1399.

Huang S, Wang F, Elliott EM, Zhu F, Zhu W, Koba K, Yu Z, Hobbie EA, Michalski G, Kang R, Wang A, Zhu J, Fu S, Fang Y. 2020. Multiyear measurements on D17O of stream nitrate indicate high nitrate production in a temperate forest. Environmental Science & Technology, 54: 4231-4239.

Li, Z., Walters, W.W., Hastings, M.G., Song, L., Huan, S., Zhu, F., Dongwei, L., Guitao, S., Li, Y., Fang, Y. Atmospheric nitrate formation pathways in urban and rural atmosphere of Northeast Chain: Implications for complicated anthropogenic effects. Environmental Pollution, 296, 2022. https://doi.org/10.1016/j.envpol.2021.118752

Lovett GM, Traynor MM, Pouyat RV, Carreiro MM, Zhu W-X, Baxter JW. 2000. Atmospheric deposition to oak forests along an urban−rural gradient. Environmental Science & Technology 34: 4294-4300.

Michalski G, Scott Z, Kabiling M, Thiemens MH. 2003. First measurements and modeling of D$^{17}$O in atmospheric nitrate. Geophysical Research Letters 30.

Nelson, D.M., Tsunogai, U., Ding, D., Ohyama, T., Komatsu, D.D., Nakagawa, F., Noguchi, I., Yamaguchi, T. Triple oxygen isotopes indicate urbanization affects sources of nitrate in wet and dry atmospheric deposition. Atmospheric Chemistry and Physics, 18, 6381-6392, 2018. https://doi.org/10.5194/acp-18-6381-2018

Xia X, Li S, Wang F, Zhang S, Fang Y, Li J, Michalski G, Zhang L. 2019. Triple oxygen isotopic evidence for atmospheric nitrate and its application in source identification for river systems in the qinghai-tibetan plateau. Science of the Total Environment, 688: 270-280.

---

## Author Comment (AC2)

*Author replies are indented and italicized.*

**Reviewer 2:**
General comments

The manuscript "Downpour Dynamics: Outsized impacts of storm events on unprocessed atmospheric nitrate export in an urban watershed" by Bostic et al. expands on the existing literature using triple isotopes of nitrate to partition storm event loads into atmospheric and terrestrial fractions. The main finding is that stormflow exports more atmospheric nitrate and baseflow exports more terrestrial nitrate in an urban watershed and there is not much difference in a non-urban watershed. This is not necessarily a surprising finding, but the results are described well and presented with interesting and unique figures with the data to support them. I think the strength of the paper is in its comparative nature. While a few others have used D17O to partition loads into terrestrial and atmospheric fractions, it is exciting to see how these relative partitions vary between comparable urban and non-urban watersheds. I think this paper would benefit from a few more citations particularly in the methods section to further differentiate it from other work in the field.

Specific Comments

Lines 39-42: I don't know if it is still correct to say that export is rarely partitioned into atmospheric and terrestrial sources. It is an important part of the literature, and this study adds to the cumulative knowledge in this area, but I wouldn't frame it as something no one else has done.

> *We agree that many prior studies have partitioned nitrate into atmospheric and terrestrial sources. Our intention was to highlight that such partitioning is less commonly done during storm events. We have revised the sentence as follows:*
>
> > *"Exported loads of individual NO3 sources (e.g., atmospheric NO3-) are less often quantified during storm events than routine baseflow samples, however (Divers et al., 2014; Sabo et al., 2016)."*

Lines 59-60: I would caution against referring to D17O as triple oxygen isotopes. While three isotopes are relevant to the measurement of D17O so the method is sometimes known as triple oxygen isotope analysis of nitrate (Kaiser 2007), but the resulting value for D17O itself is not triple. D17O values could also be called the 17O anomaly (Michalski 2003). To add to the confusion, the way you are using these isotopes for quantifying loads is often referred to as triple nitrate isotopes, where D17O is one of the three isotopes along with d15N and d18O (Liu et al 2013, Hale et al 2014, Rose et al 2015). To minimize

confusion here and throughout the paper, I would keep it as either "D17O values" or "triple oxygen isotope analysis."

> *Thank you for this suggestion. To reduce potential confusion, we will replace the two instances of "triple oxygen isotopes" in the manuscript with "oxygen isotopes" and then use the abbreviation of "D17O" in the remainder of the manuscript.*

Line 79: What are "moderate frequency samples"? There is no reference point for the time interval.

> *Thanks for this comment. The sentence has been changed to:*
>
> > *"To address these research questions, we collected moderate-frequency (45 minute – 12 hour) streamwater samples before, during, and after eight rainfall events, bulk rainfall samples corresponding to these events, as well as monthly baseflow samples, in two catchments within the broader Chesapeake Bay watershed."*

Line 101: 45 minutes to 12 hours is a very wide range of sampling intervals. Is this the average among events with widely different sampling intervals, or does this change within a given event?

> *The sampling frequency did sometimes change within a given event. For clarification, we will add a supplementary table that includes relevant information (sample date, time, discharge, nitrate concentrations and isotopes, water isotopes) to your question. Sampling intervals were generally shorter during the beginning of an event (i.e., the rising limb) and the interval was longer later in the event (i.e., the falling limb). The longest sampling interval (12 hours) was associated with the slowly falling limb of a large event.*

Lines 166 – 170: I am not sure what traditional methods you are referring to. Maybe a citation or two would help. The other papers I have seen that quantify NO3 loads use the discharge that corresponds with each individual grab sample (ie Hale 2014).

> *Thanks for your suggestion to clarify the methods we are referring to. We have changed the sentence to:*
>
> *Line from our paper to edit: To assess potential bias in $NO_3^-{}_{Atm}$ load quantification between our method (i.e., multiple samples collected during a storm event; eq. 5) and methods in which a single sample is collected, we used the mean daily discharge multiplied by $NO_3^-{}_{Atm}$ concentrations of each individual grab sample collected during a particular event.*

Lines 222-223: The fractional export plot seem like a very interesting method. Could you add a few citations for other that have used this, unless you are the first?

*After a literature search, we are unaware of others who have used a scatter plot in this same method. Just because we are unaware we do not claim to be the first, however.*

Figure 2: I would recommend using open and filled circles/triangle for your two sites. With the colors as they are, they will both print grey in black and white.  Also, is each point a single storm event? It is a bit confusing to have both NO3 atm and NO3 terrestrial plotted as they are directly inverse of each other.

[Figure]

*Thanks for your suggestion. We have changed this figure so that circles are filled and triangles are open. Each point is a single storm event. We included both NO3-Atm and NO3-Terr to show the differences in export behavior.*

Figure 3: Event mean does gloss over the changes in source load within a given event. Though I suppose it is necessary for the 1:1 comparison with baseflow for this plot. Might be worth discussing in the body of the text though.

*Figure 3 was produced to demonstrate the changes between baseflow and event mean concentrations/event mean values. This manuscript focused on event mean*

*concentrations/values and loads relative to baseflow as opposed to intra-storm variation.*

Line 16:  Spell out 8

*"8" is now spelled out as "eight"*